# Electrocorticography reveals thalamic control of cortical dynamics following traumatic brain injury

Sima Mofakham [1✉], Adam Fry[1,2], Joseph Adachi[1], Patricia L. Stefancin[3], Tim Q. Duong[3], Jordan R. Saadon[1], Nathan J. Winans[1], Himanshu Sharma[1], Guanchao Feng[4], Petar M. Djuric[4] & Charles B. Mikell [1✉]

The return of consciousness after traumatic brain injury (TBI) is associated with restoring complex cortical dynamics; however, it is unclear what interactions govern these complex dynamics. Here, we set out to uncover the mechanism underlying the return of consciousness by measuring local field potentials (LFP) using invasive electrophysiological recordings in patients recovering from TBI. We found that injury to the thalamus, and its efferent projections, on MRI were associated with repetitive and low complexity LFP signals from a highly structured phase space, resembling a low-dimensional ring attractor. But *why do thalamic injuries in TBI patients result in a cortical attractor*? We built a simplified thalamocortical model, which connotes that thalamic input facilitates the formation of cortical ensembles required for the return of cognitive function and the *content* of consciousness. These observations collectively support the view that thalamic input to the cortex enables rich cortical dynamics associated with consciousness.

[1] Department of Neurosurgery, Renaissance School of Medicine at Stony Brook University, Stony Brook, NY, USA. [2] Department of Rehabilitation and Human Performance, Icahn School of Medicine at Mount Sinai, New York, NY, USA. [3] Department of Radiology, Renaissance School of Medicine at Stony Brook University, Stony Brook, NY, USA. [4] Department of Electrical and Computer Engineering, Stony Brook University, Stony Brook, NY, USA.
✉email: Sima.Mofakham@stonybrookmedicine.edu; Charles.Mikell@stonybrookmedicine.edu

Fundamental mechanisms of consciousness are mysterious, despite significant recent theoretical and experimental progress. Thus, the return of consciousness following TBI is uncertain and hard to predict[1,2]. While corticothalamic circuits are believed to be crucial, few direct observations of their function are available. Most studies of cortical signals have used electroencephalography (EEG)[3–6], which is hampered by limited spatial and frequency resolution. However, the features of EEG associated with consciousness are now reasonably well-characterized; EEG in conscious subjects has greater spectral power in higher frequencies[7,8], increased complexity[4,9,10], and higher information content[5] relative to EEG recorded in unconscious subjects. However, how these features arise is not known with certainty due to a lack of direct observations from involved regions and a lack of lesion-type experiments needed to make causal inferences.

Studies of severe traumatic brain injury (sTBI) represent an opportunity to address both gaps and a clinical need. Injury to the cortex, subcortical white matter, and thalamus characterize sTBI[11,12], and serial MRI studies have reported extensive brain atrophy after sTBI[13–15]. Thalamic atrophy is strongly correlated with prolonged unconsciousness, leading to the question of how thalamic function facilitates consciousness[16,17]. Moreover, the integrity of frontothalamic circuitry also correlates with the level of consciousness[18,19], which highlights the critical need to understand the role of the thalamus in consciousness.

Early studies in brain slices and intact animals have led to the longstanding view that the firing mode of the thalamus correlates with the level of consciousness[20,21]. It has been argued that rhythmic oscillations in the somatosensory thalamus are not conducive to the transmission of complex information[22]. This observation led to the proposal that thalamic neurons control the cortical transition from sleep to wakeful states and facilitate information transmission[23]. Investigations in the 2010s have revealed functional significance to this transition; thalamic input has been shown to synchronize cortical oscillations facilitating cortico−cortical information transmission in the visual system[24]. In the frontal lobe, the mediodorsal thalamus functions as a content-free amplifier of cortical representations that boosts synaptic gain and thus enables neuronal ensemble formation in the prefrontal cortex (PFC)[25,26]. These data are consistent with a model in which the thalamus capacitates cortical representations needed for consciousness.

However, it is not clear whether the thalamus's key function in the recovery of brain injury is the maintenance of arousal (especially from central thalamic efferents[27]) and/or facilitation of cortical function leading to complex behavior[28]. According to two recent reports, thalamic injury was not strongly associated with unconsciousness[29,30], and there is no apparent impairment of arousal with pure thalamic injuries[29]. However, stimulation of the central thalamic nuclei does appear to increase the level of consciousness in several reports, in both chronic[31] and subacute[32] traumatic brain injury. Given the massive burden of sTBI worldwide, a detailed understanding of the contribution of thalamic activity to the recovery of consciousness is critical.

Here, we use direct recordings of local field potentials (LFP) from frontal areas implicated in consciousness to understand how neural signals associated with consciousness arise. We had a rare and unique clinical opportunity: we recorded and stimulated depth electrodes implanted in the PFC and anterior cingulate cortex (ACC) for seizure monitoring after sTBI. Due to the current limitations for direct recording from the thalamus in sTBI patients, we used *in silico* modeling to gain insight into the role of the thalamus in shaping the functional state of the cortex in the context of recovery of consciousness. Our model suggests that thalamocortical projections to the frontoparietal network (FPN) facilitate the complex dynamics needed for consciousness. By contrast, injury to these connections results in a dysfunctional state of cortical networks, incapable of maintaining neuronal ensembles required for consciousness.

## Results

**Study population.** Five sTBI patients were enrolled in the study, and we obtained combined LFP/EEG and recorded responses to single-pulse electrical stimulation from all five (see "Methods"). We also enrolled a single control patient with epilepsy. There were no surgical complications in the patients, and the study procedures were tolerated well. Ultimately, four of the five patients returned to consciousness over days to weeks (Table 1). In one case, a patient was comatose at implant and almost fully conscious by the time the study procedures were conducted (Subject 4, CRS 10, Table 1; this patient was explanted rapidly). We implanted a single depth electrode spanning ACC and PFC (Fig. 1a) based on prior data supporting frontal depth electrodes in comatose patients for seizure monitoring[33]. Complete patient histories are found in Supplementary Note 1. See Supplementary Fig. 1 for a detailed timeline of injury, implantation, and measures of consciousness.

**Higher frequencies in LFP are associated with consciousness.** We recorded depth LFP from PFC and ACC in all study subjects. Figure 1b shows these recordings in a recovering comatose patient (blue) and a patient who never recovered consciousness (red). Consistent with prior reports, we found that unconsciousness was associated with delta-band oscillations in ACC and PFC (Fig. 1c, red). The recovery of consciousness is associated with the development of faster theta, alpha, and beta

### Table 1 Patient summaries.

| | Subject 1 | Subject 2 | Subject 3 | Subject 4 | Subject 5 | Control |
|---|---|---|---|---|---|---|
| Age | 78 | 54 | 48 | 25 | 38 | 24 |
| Sex | M | M | M | M | M | F |
| Mechanism | 10 ft fall onto concrete | Moped accident without helmet | Pedestrian struck by vehicle | Motorcycle accident | Cyclist without helmet struck by car | — |
| Side of electrode | Right frontal | Left frontal | Right frontal | Right frontal | Right frontal | Bilateral |
| Injuries to thalamocortical loop | No | No | No | Yes | Yes | No |
| Initial GCS | 6* | 5 | 3 | 3 | 4 | 15 |
| CRS-R (initial-final) | 2—10 | 0—7 | 0—2 | 10 | 0—2 | 23 |
| Time until command following | 7 Days | 38 Days | 51 Days | 5 Days | N/A, expired after 196 days | — |

Abbreviations: *GCS* Glasgow Coma Scale, *CRS-R* Coma Recovery Scale-Revised.
*Patient initially presented with a higher GCS (14) but declined prior to enrollment.

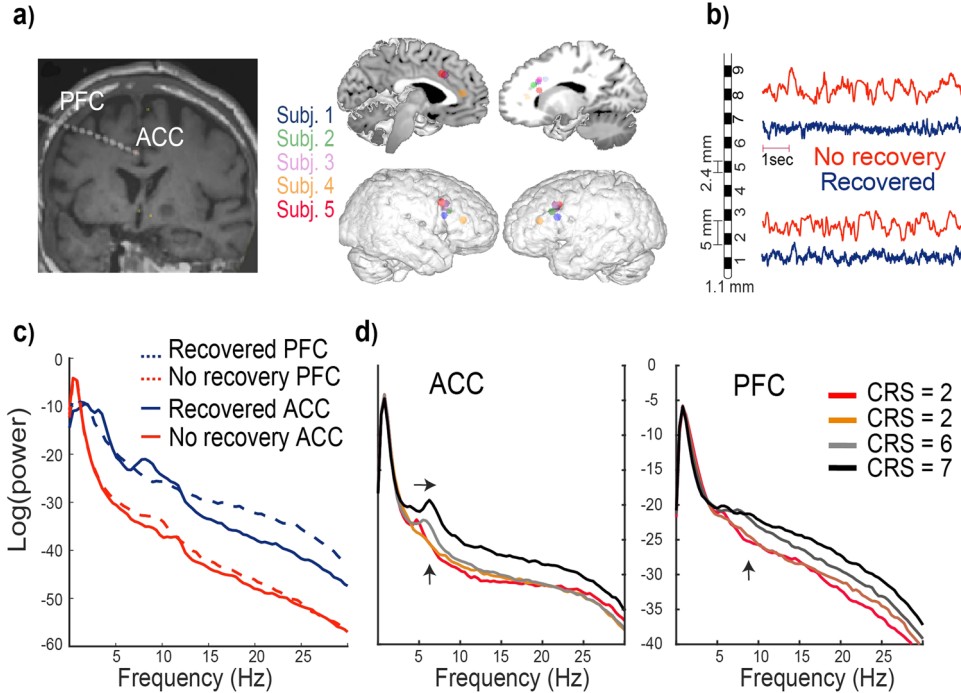

**Fig. 1 LFP activity distinguishes coma, wakefulness, and the transition from one state to another. a** Left: a depth electrode is placed to provide direct recordings throughout PFC and ACC. The depth electrode location is shown on a T1-weighted coronal MRI. All electrodes were right-sided, except for Subject 2, whose electrode was on the left. Right: locations of electrodes from external and internal sagittal slices for all the patients. **b, c** Representative LFP recordings and their associated power spectral density recorded from PFC and ACC from the patient who recovered (Subject 1, CRS-R-2—10) versus the deeply comatose patient who did not recover (Subject 5, CRS-R-0-2). Note that the wakefulness exhibits spatially distinctive peaks in the ACC (theta and alpha) and PFC (delta and beta). **d** A patient who recovered consciousness over days of recordings (Subject 2) showed a gradual shift of the ACC theta peak towards higher frequencies and increased beta power in the PFC (black arrows).

frequencies (Fig. 1c, blue). We observed increases in theta-band oscillations in ACC (Fig. 1d, left) during the return of consciousness and beta-band oscillations in PFC (Fig. 1d, right). Power spectra for all subjects over days of recordings are included in Supplementary Fig. 2.

**Complexity and variability of cortico−cortical evoked potentials are associated with consciousness.** Consciousness has previously been associated with the complexity of the brain's response to perturbation reflective of its capacity for information integration (measured as the "perturbational complexity index" [PCI] in prior reports)[4,34]. Thus, we applied single pulses of electrical stimulation and recorded cortico−cortical evoked potentials (CCEP; Fig. 2a, b). In general, ACC stimulation resulted in a broader spatial response than PFC stimulation (Supplementary Fig. 3b), suggesting that ACC has widespread connectivity to other cortical areas. Thus, most single-pulse stimulation in this study was administered to ACC. Patients with strictly cortical and/or white matter injuries exhibited characteristic multiphasic responses to stimulation with a high level of complexity reflected in increased zero crossings (Fig. 2d, left panel, $P < 0.01$, two-tailed t-Test) and Lempel−Ziv complexity (Fig. 2d, right panel, $P < 0.01$, two-tailed t-Test) as well as increased variability of the CCEP across trials. However, the most gravely injured patient with a bilateral thalamic injury had a stereotypical monophasic response (Fig. 2c, red line at bottom), which was broadly detectable throughout the cortex (Fig. 2a, red). These features were similar at longer timescales (tens of seconds).

Single pulses of stimulation revealed a high degree of susceptibility to external perturbations in two patients with thalamic injuries. This observation was reproducible over days. The most severely injured patient with bilateral thalamic injury

exhibited a high degree of susceptibility to single pulses of stimulation. In this patient (Subject 5) and the patient (Subject 4) with unilateral thalamic injury, trains of single pulses of stimulation produced highly deterministic cortical responses (Supplementary Fig. 4). However, trains of stimulation failed to produce deterministic responses in patients with strictly cortical injuries who quickly recovered consciousness. These observations suggest a modulatory role for the thalamic input to the cortex.

We note that one patient who rapidly recovered consciousness (Subject 4) exhibited features similar to the most gravely injured patient. Both patients had evidence of injury to the thalamus and/ or its efferent fibers; in the case of the patient who rapidly recovered, the injury was unilateral (Supplementary Fig. 5). These observations led us to assess the mechanistic role of the thalamus.

**Thalamic injury leads to a low-dimensional, ring attractor-like state in vivo.** Phase space analysis is a parsimonious representation of how complex systems change over time, especially when these systems are nonstationary or otherwise difficult to represent in analytic terms. Previous dynamical analysis of wakeful EEG has identified a transition between two attractors driven by thalamic input[35]. We, therefore, hypothesized that a lack of thalamic input would constrain cortical dynamics during coma. To understand the underlying dynamics of coma, we reconstructed the phase space of PFC recordings using delay embedding. We used autocorrelation and false nearest neighbors (FNN) metrics to estimate the appropriate time lag and the embedding dimension[36].

We reconstructed the two-dimensional phase space of ten seconds of PFC dynamics (taken from five minutes of LFP recording) for all five patients (Fig. 3a). The phase space of cortical dynamics for patients with thalamic injuries is structured

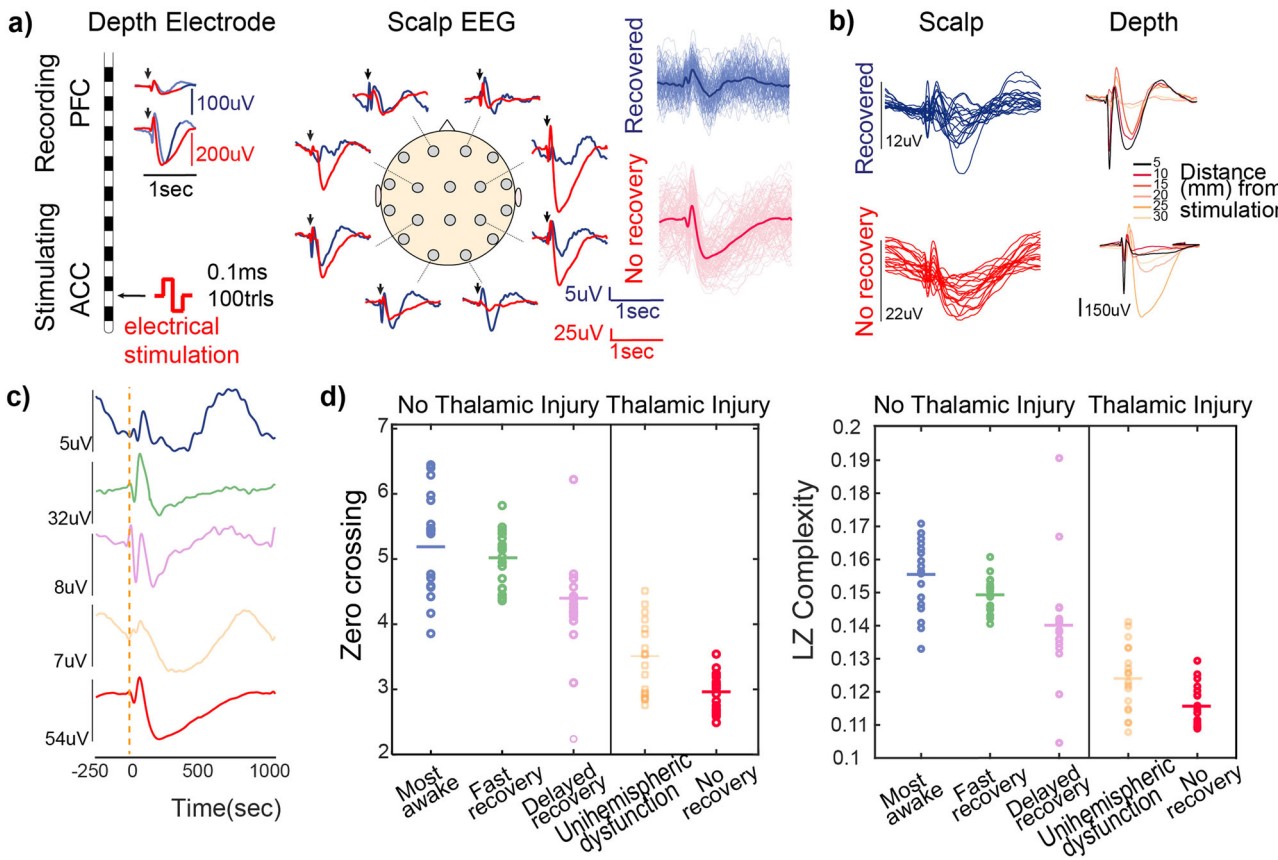

**Fig. 2 Cortical evoked potentials exhibit distinctive features in patients with thalamic injury. a** The evoked responses to ACC stimulation in a recovering comatose patient with no thalamic injury and a patient with a bilateral thalamic injury who never recovered consciousness are overlaid in blue and red, respectively. **b** Propagation of these evoked responses across scalp and depth channels are shown. **c** A representative scalp evoked response to ACC stimulation in each sTBI patient is shown. **d** Evoked responses across patients were classified based on their complexity using zero-crossing (left panel) and LZ complexity (right panel). Each circle represents the average of evoked recordings across trials (90−100 trials per patient) from each scalp contact. Horizontal lines represent the average across all scalp contacts. Evoked responses from patients with thalamic injury exhibited significantly diminished complexity with both measures ($n = 3564$ trials recorded from 18 scalp contacts in patients with thalamic injury and 5220 trials for patients without thalamic injury, $P < 0.001$, two-tailed t-Test). In this figure, "fast recovery" and "delayed recovery" refer to Subjects 2 and 3, who followed commands at 38 and 51 days, respectively.

with a small repertoire of available states that are periodically visited, resembling a limit cycle attractor in the phase space (Fig. 3a, top right panels, Subjects 4 and 5). Also, PCA on these recordings from depth contacts and scalp EEG contacts showed that fewer dimensions were required to represent their data (Supplementary Fig. 6). In contrast, patients without thalamic injuries required a larger number of dimensions to explain the variance in the data (Supplementary Fig. 6) and no attractor-like structure was evident in the phase space (Fig. 3a).

We examined the robustness of the attractor structure observed in the phase space by perturbation (single-pulse electrical stimulation at 1 Hz). Figure 3b shows how a ten-second train of single pulses of stimulation affects the temporal evolution of the visited states in phase space. The gray trajectories are associated with ten seconds of PFC LFP before stimulation. The red trajectory represents the ten-second period when single pulses of stimulation at 1 Hz were applied. In patients with thalamic injury (Fig. 3b, right panels), the system repetitively and predictably visits the same states (notice similar-appearing straight red lines). When stimulation was applied to the patients without thalamic injury, the response to stimulation was much less predictable.

To summarize, we found that thalamic injuries put the cortical networks in a dysfunctional state with simple, low variability and repetitive dynamics. This functionally

passive regime has a limited number of allowed states with periodic trajectories between these allowed states, which resembles a low-dimensional attractor stable to external perturbation/stimulation. We then sought to identify the mechanism underlying the attractor-like dynamics in patients with thalamic injuries.

**Thalamic injury recapitulates delta-band, low-complexity LFP activity *in silico*.** To evaluate how the thalamic input to FPN could explain the LFP features we observed in coma, we have built an abstract model of FPN structures *in silico* using a simplified network of leaky-integrate-and-fire (LIF) neurons[37,38]. Although this abstract model lacks many of the biological details of the thalamo-cortical circuit, it still replicates many experimental observations and makes valuable predictions. This model includes two populations representing cortical cells: one in the frontal cortex and another in the parietal cortex. Each receives inputs from a thalamic population. Each group of cortical neurons is recurrently connected, and the two populations also communicate with each other via long-distance connections. Thalamic cells do not directly interact with each other but instead project to the prefrontal and parietal cortices (Fig. 4a). We posited two linked roles for the thalamus in this network: (1) *driving*: providing *excitatory* input to their post-synaptic neurons, and (2)

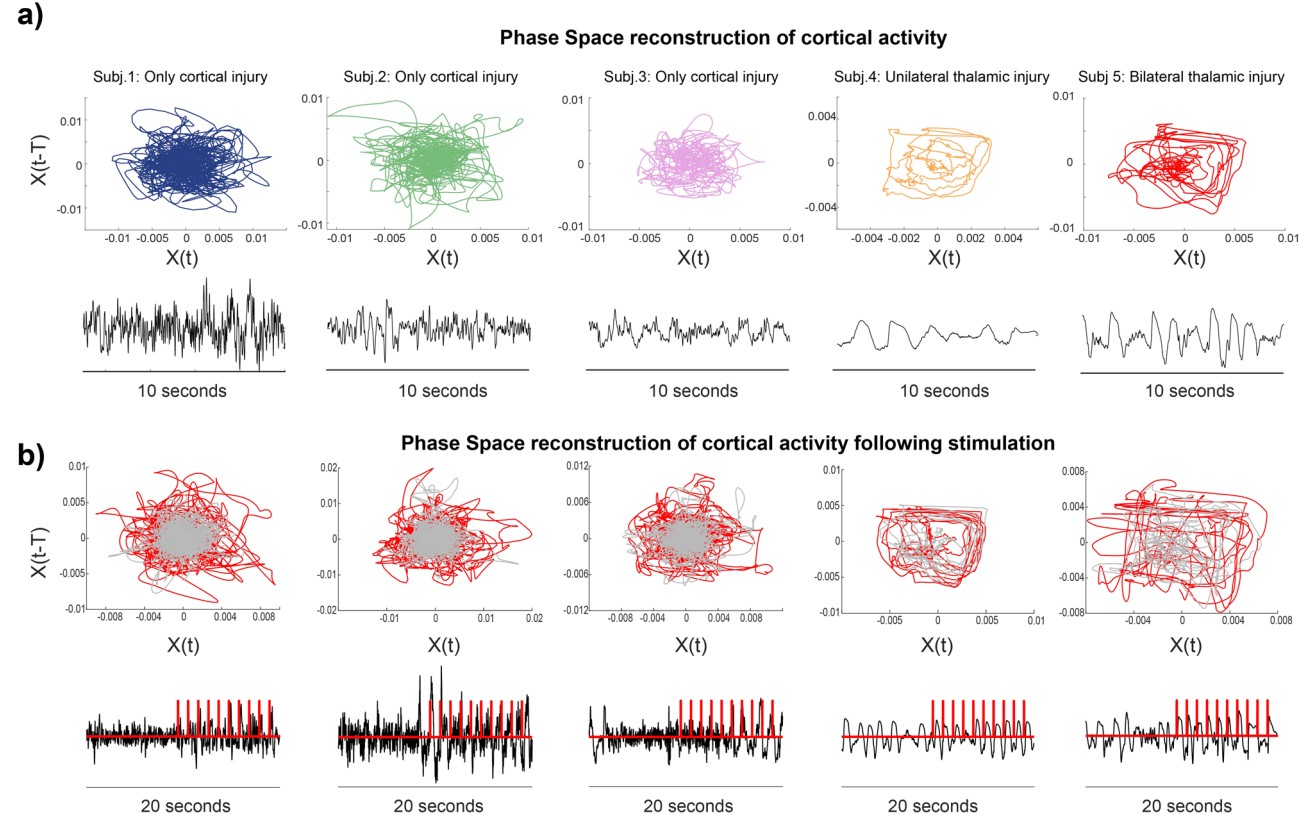

**Fig. 3 Reconstructed phase space of depth cortical recordings reveals a low dimensional attractor for patients with thalamic injury. a** We reconstructed the phase space of recordings obtained from depth PFC recording for all the patients. Patients 4 and 5 with unilateral and bilateral thalamic injury, respectively, show a structured phase space that repeatedly visits similar trajectories reflecting a limited number of available states in the repertoire of cortical networks. **b** Phase space of 20 s of cortical recordings consisted of a ten-second baseline (gray) followed by a ten-second train of single pulses of stimulation (red). The timing of single pulses is shown in red on the recorded LFP. Responses are consistent with a ring attractor in patients 4 and 5.

*modulatory*: adjusting the cortico−cortical *connectivity* in which thalamic activity amplifies the strength of connectivity across ($\beta$) and within ($\alpha$) cortical networks[25,26]. Here, we simulated coma as a state where the thalamic activity is reduced and phasic, similar to sleep[20]. In the awake state, the thalamus exhibits a tonically high firing rate in diverse spatiotemporal patterns (Fig. 4b)[21].

When we reduced the thalamic activity in the model, we found that the cortical activity is also reduced, and the overall simulated cortical baseline LFP activity shifted towards the delta-band (Fig. 4b). Moreover, single-pulse electrical stimulation of the model in a coma state (reduced thalamic activity) resulted in simple, broad, and monophasic responses (Fig. 4c) similar to what was experimentally observed when the thalamic input was missing (Fig. 2a, c, and d).

**Thalamic injury leads to a low-dimensional ring attractor-like state *in silico*.** We reconstructed the phase space of the simulated coma and awake states using a similar approach as in Fig. 3. We used FNN and autocorrelation to estimate the appropriate delay and embedding dimension and plotted the state trajectories over time. The phase space was much more complex and random when thalamic input was fully present (awake state), (Fig. 4b, left, bottom row). When thalamic input was reduced (coma state), the phase space was simple and structured (Fig. 4b, right, bottom row), resembling a ring attractor, in which cortical networks are only allowed to visit a limited number of states repetitively. Together, these data support the view that the thalamic input allows the cortical networks to visit a larger repertoire of available states with a higher degree of freedom of transitioning from one to the other.

**Thalamic injury impairs neuronal ensemble formation *in silico*.** Recovery of consciousness requires both arousal and goal-directed behavior[39]. Goal-directed behavior, in turn, requires activation of neural ensembles encoding specific behaviors[40]. With our *in silico* model, we set out to understand how activation of these neural ensembles is impaired in coma. We embedded a neuronal ensemble in the model by strengthening synaptic weights among a subpopulation of PFC neurons to form a clique of neurons that could support a thought, memory, or another functional subcomponent of behavior. When there is no thalamic amplification of cortical connections (as in the simulated coma state), the activity of such a cluster of neurons follows the ongoing cortical "up" and "down" states similar to the rest of the neurons that are not in the cluster (Fig. 4d, right raster plot). However, in the awake state, an embedded pattern reemerges due to the thalamic strengthening of neuronal functional connectivity. This pattern exhibits distinct and stable firing patterns that differentiate this neural subpopulation from the rest of the network (Fig. 4d, left raster plot). These data are consistent with the hypothesis that thalamic input needs to increase excitability and both local and long-range connectivity to optimize cortical function as measured by ensemble activation.

**Discussion**
The return of consciousness after TBI is associated with a well-described transition from delta-band to alpha, beta, and theta oscillations. We extend these findings by reporting that thalamic injury is associated with an attractor-like dysfunctional cortical state characterized by simple and repetitive ECoG activity in vivo and in silico. Single-pulse electrical stimulation of the cortex

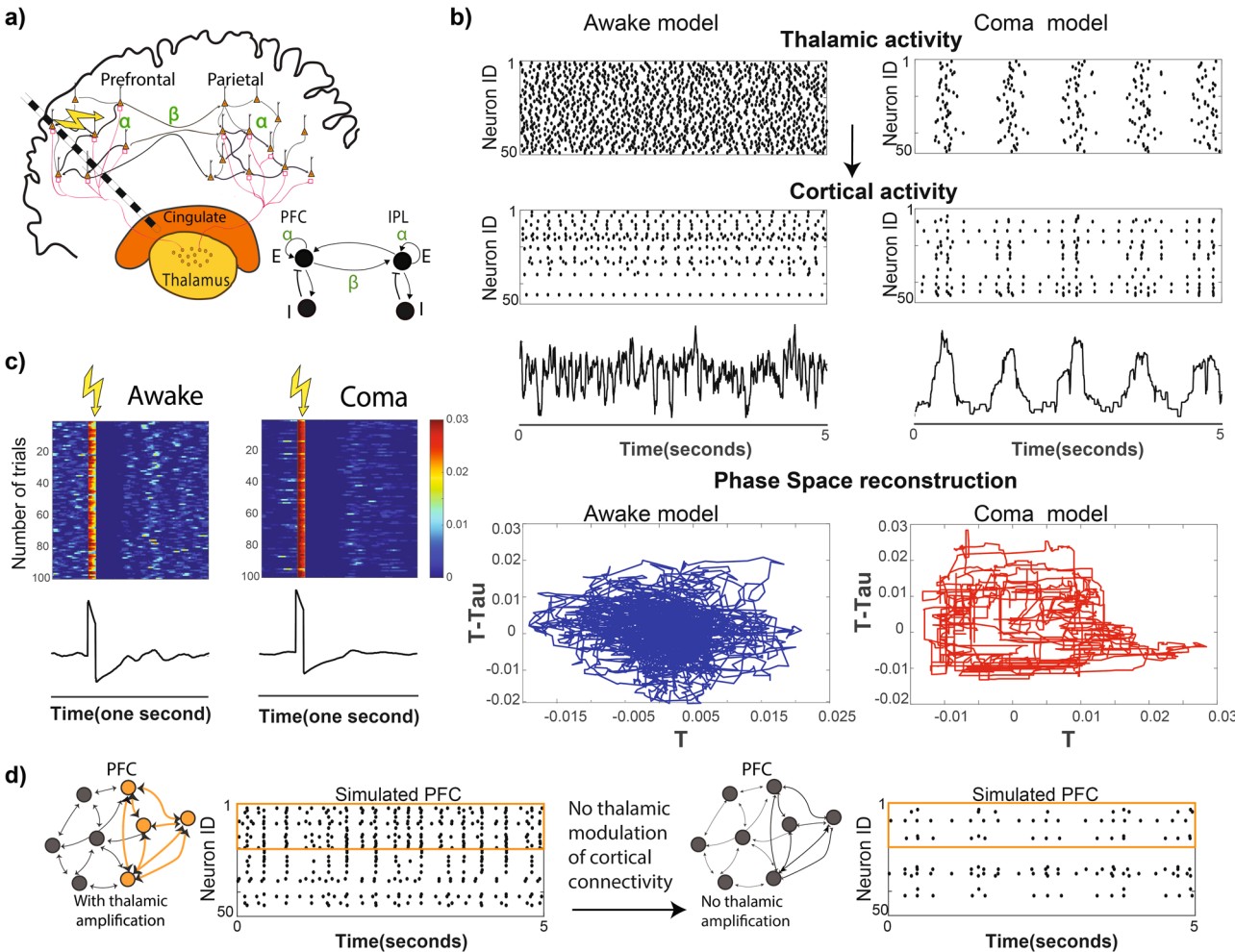

**Fig. 4 The computational model of thalamocortical networks in consciousness predicts *content* of consciousness requires thalamic modulation of cortical networks. a** The thalamocortical model consists of two interconnected cortical populations that receive thalamic input. Each cortical region includes interacting excitatory and inhibitory cells. In addition to the excitatory input, the thalamus modulates the local ($\alpha$) and long-distance ($\beta$) connectivity of cortical networks. **b** Depending on the activity mode of thalamus (tonic vs. phasic) cortical cells switch between rhythmic (coma) and asynchronous (awake) dynamics. The reconstruction of phase space for awake and coma shows a structured phase space for coma and much more complex phase space for the awake state; similar features are seen in the experimental data. **c** The evoked responses for 100 simulated trials of coma and awake states are shown in heat maps for a one-second period where the color intensity denotes the deflection from the baseline. Underneath the heat map, the average of these responses across one hundred trials is shown; the stimulation time is marked by the yellow lightning bolt. Note that stimulation in the simulated awake state triggers multiple waves of excitation. Stimulation in coma results in a monophasic, and stereotypical response. **d** In our model, an embedded cortical cluster (i.e., an ensemble, orange bracket on raster plots) only becomes activated when it receives both excitatory thalamic input and amplification of its cortico—cortical connections (left versus right raster plots). This ensemble represents a low-level "building block" of consciousness.

generated monophasic, repetitive, deterministic evoked potentials when thalamic injury was present. We replicated these findings in silico; our hypothesis-driven model had strong homology to the recorded electrocorticography. Moreover, in our model, thalamic input facilitated the formation of neuronal ensembles required for the *content* of consciousness; neuronal ensembles did not form when the thalamic input was reduced. We suggest that simple and repetitive cortical dynamics associated with the coma attractor (when thalamic input is absent) are inadequate to engage the vast repertoire of ensembles needed for goal-directed behavior consistent with other recent models[41]. This leads to a limited behavioral repertoire of the comatose sTBI patient. Critically, the thalamus has to adjust excitability and connectivity to bring about complex dynamics, consistent with recently proposed theoretical models of thalamic function[42].

To the best of our knowledge, this study represents the first report combining electrocorticography with neuroimaging to understand how thalamic injury affects cortical function. These data have important implications for coma therapeutics. Specifically, this model furnishes an explanation for the mixed results of central thalamic deep brain stimulation in TBI patients[43]. Our data imply that when higher-order thalamic projections are injured, interventions to increase cortical excitability alone are unlikely to restore consciousness because the full repertoire of human behavior requires dynamic adjustment of connectivity. Further work will be required to understand how central thalamic stimulation affects connectivity; it has clear effects on cortical excitability[44], but deficits in synaptic connectivity may explain why it has mixed results in the clinical treatment of consciousness disorders[31,43]. It is possible that stimulation of mediodorsal or other thalamic nuclei could increase synaptic connectivity in the required fashion. Finally, our data provide a possible explanation for recent observations that thalamic injuries do not totally prevent consciousness[29]. Based on our model, we propose that

thalamic injury leads to impairment of neuronal ensemble formation in the cortex, and thus, the *content* of consciousness; thus, even extensive thalamic injuries could allow for a limited repertoire of conscious content (i.e., the cortical ensembles that can be activated by the intact thalamic projections). In the case of Subject 4, he was conscious, despite an extensive but partial thalamic injury. A detailed neuropsychological examination may have revealed deficits related to right hemisphere dysfunction, but the language-based control needed to assay consciousness was evidently intact. When there is a bilateral thalamic injury, there is likely to be less functioning cortex needed to support consciousness. New therapies for brain injury should focus on restoring the thalamic role in modulating cortical networks and thus the *content* of consciousness, in addition to its role in augmenting arousal.

Conscious behavior exhibits similarities with the low-predictability dynamics we identified in the conscious LFP recordings. William James described that thalamic input to the cortex in frogs leads to unpredictable behavior; in contrast to a frog with the thalamus ablated, "…[the frog's] conduct has become incalculable. We can no longer foretell it exactly. The effort to escape is his dominant reaction, but he *may* do anything else…"[45]. After a severe brain injury, the return of consciousness is similarly characterized by a gradual return of increasingly complex goal-directed behavior[46]. While increased complexity of EEG in conscious patients has been previously reported[4,6], our data also show that the return of consciousness implies decreased predictability as well as increased complexity. This lack of predictability is a consequence of thalamic input to the cortex, which is in turn intimately linked with the cortex's ability to form neuronal ensembles.

Further research is needed to define the dynamics of consciousness exactly; a tantalizing possibility is that the dynamics are formally chaotic[47], consistent with the low predictability of animal behavior observed by James. However, it is currently not possible to distinguish this possibility from simpler kinds of nonlinear interactions[48]. We can currently only conclude that the conscious cortex exhibits complex dynamics with low predictability and a large number of possible states.

Our sample size is relatively small ($n = 5$) and limited to a single disease process. Other conditions such as cardiac arrest may lead to coma in a manner distinct from TBI. We also did not directly observe electrical activity in the thalamus because we do not routinely place electrodes in this structure for seizure monitoring. Even so, some groups have advocated thalamic monitoring for seizures in refractory epilepsy cases[49]. However, in the case of critically-ill, comatose sTBI patients, it could be beneficial to implant thalamic electrodes—especially if preclinical data suggest that stimulation could be therapeutic. Animal models will likely be only modestly informative about the role of the thalamus in disorders of consciousness because of the challenges associated with studying consciousness in animals as well as the difficulty that accompanies modeling traumatic coma.

In addition, although our computational model is a simplified model and lacks many of the biological details of the thalamo-cortical circuit, it replicates many of our in vivo observations and makes important predictions. Further experiments are needed to refine the model in ways that reflect emerging evidence about the organization of these circuits, and will shed additional light on how they recover from injury.

## Methods

**Patient selection**. One control subject (aged 24) and five patients (aged 25−78) who became comatose (GCS < 8) after suffering severe traumatic brain injury were recruited for this study after admission to Stony Brook University Hospital from 2016-2018. Patient clinical histories are summarized in Supplementary Note 1.

Candidate sTBI patients were identified during neurosurgical consultation in the emergency department. A seizure monitoring stereotactic depth electrode was placed to detect seizures; in one recent series, seizures or periodic discharges occurred in 61% of sTBI patients, and 43% of these seizures were observable only with depth recordings[50]. Informed consent to enroll in the study and implant depth electrodes was obtained from the patient's Legally Authorized Representative (LAR) according to NIH, Declaration of Helsinki, and institutional guidelines, and was monitored by the Stony Brook University Hospital Committee on Research Involving Human Subjects, as well as a separate Data Safety and Monitoring Board (DSMB). Pregnant females and patients with major brain structural abnormalities (excepting hemicraniectomy) that would preclude between-subject comparisons were excluded from the study (see Supplementary Table 1 for full inclusion and exclusion criteria). Supplementary Fig. 1 describes the exact timeline of injury, depth electrode implantation, and CRS-R scoring. Supplementary Note 1 provides additional clinical information.

**Electrophysiological recordings and stimulation**. Patients were implanted with a ten-contact stereotactic depth electrode (Ad-Tech, Spencer Depth™) spanning the dorsolateral prefrontal cortex (DLPFC) to ACC. Electrodes were right-sided for all patients except for Subject 2 (see Supplementary Note 1). These areas were selected for the study because it is technically simple to place depth electrodes there for clinical seizure monitoring. FPN is well-known to be critical to goal-directed behavior[51]; ACC is also a well-defined "hub" region that has broad cortical connectivity[52]. We thus expected to probe cortical function broadly from ACC as a single point of access. PFC was identified as the caudal part of the middle frontal gyrus at the level of the coronal suture, and the anterior cingulate was typically targeted 20 mm from the tip of the frontal horn. Each contact can be used for ECoG recordings and stimulation. We also attached scalp EEG leads in an 18-contact standard 10−20 montage. Impedances for scalp contacts were within 5−10 kΩ. Impedance was not directly measured on intracranial contacts, but recordings were visually inspected for quality. Artifacts were rejected by visual inspection as well as by the Fieldtrip Toolbox[53]. After implantation, patients underwent detailed behavioral testing, stimulation, and recording. After implantation of the depth electrode, continuous scalp (EEG) and depth (ECoG) monitoring was initiated to detect seizure activity and to provide antiepileptic treatment and prophylaxis as needed. The sedative medication for each patient is listed in Supplementary Table 2.

Single-pulse stimulation (10 mA, 100 μA, 100 stimuli) was applied to contacts within the ACC and DLPFC using a Nicolet cortical stimulator (Natus Medical: Pleasanton, CA) while recording all other contacts including scalp and depth electrodes[54]. Stimulation pulses were biphasic and bipolar and applied between two adjacent contacts. The maximum parameters were 10 mA at ≥1 Hz (biphasic square-wave pulse, 100 μS per phase); 10 mA was selected based on prior reports[55]. No adverse effects were noted from stimulation. There were no obvious changes to the level of consciousness or ongoing EEG activity as a result of single-pulse stimulation.

**Behavioral testing**. After enrollment, we measured the CRS-R score on a daily basis before delivering single pulses of stimulation in each patient to assess the level of consciousness[39]. Most patients gradually returned to consciousness over a period of days or weeks (Table 1).

**Imaging**. Imaging was obtained at multiple time points to verify proper electrode placement. CT was obtained post-operatively after implantation to localize the electrode and to identify potential surgical complications. MRI was performed after electrode removal and is shown in Supplementary Fig. 5. MRIs were obtained for clinical (not research) purposes and represent the best images that could be obtained under the circumstances. Volumetric Siemens FLAIR (fluid-attenuated inversion recovery) sequences were used for all patients. The matrix size was $256 \times 256$. Slice thickness/gap was 1/0 mm. Flip angle = 120, TE/TR = 335/5000 ms (exception: TR = 4500 for subject 1), TI = 1800 ms, pixel size = 0.9375 mm. Scans were typically acquired in the sagittal plane and resampled to axial. For Subject 4, diffusion tensor images were acquired using an EPI sequence with flip angle = 90, TE/TR = 90/5400 ms, slice thickness/gap = 4/0 mm, 32 diffusion direction, single shell scheme with $b = 1000$. For Subject 5, a diffusion-weighted EPI image is also shown (TE/TR = 90/10300 ms; flip angle = 90, $b$ = acquisition matrix = $200 \times 200$ and slice thickness/gap was 4/0 mm). Field strength was for 3T for Subjects 1, 2, and 4 and 1.5T for Subjects 3 and 5.

**Human subject safety**. All study procedures were conducted under the supervision of a Data Safety and Monitoring Board and the Stony Brook University Committee on Research in Human Subjects (CORIHS). There were no complications from depth electrode placement or stimulation in this study. In all cases, MRI after implantation did not show evidence of injured tissue in the area of the electrode. Notably, the used current amplitudes and charge densities were below the manufacturer's recommendations and are comparable to stimulation parameters used for therapeutic cortical stimulation (in some cases, ≤8 mA) chronically[56], with bursts up to 12 mA[57]. We had no study-related adverse events.

**Data analysis**. Data analysis was performed using MATLAB. Scalp and depth channel recordings were subject to mean subtraction, bipolar re-referencing, and band-pass filtering (0.5−30 Hz).

**Frequency analysis**. Baseline power spectral density (PSD) estimates shown in Fig. 1, and Supplementary Fig. 2, were created using Welch's overlapped segment averaging estimator with a Hamming window and 50% overlap. Frequency analysis was performed over the 0.5−30 Hz range.

**Cortico−cortical evoked potentials (CCEPs)**. CCEPs were created by cutting the recorded EEG and ECoG signals into one-second epochs (from 250 ms pre-stimulation to 750 ms post-stimulation). These were time-locked to the depth electrode stimulation pulse. The CCEPs recordings were band-pass filtered with cutoff frequencies of 0.5−30 Hz and then were demeaned and averaged.

**Complexity analysis**. The zero-crossing (ZC) score was used to estimate the complexity of the CCEP[58]. This measure represents the number of times that the demeaned and band-pass filtered (0.5−10 Hz, 4th order FIR filter) CCEP signal crossed zero (changed sign) per second. This measure was computed for each trial, and the averaged ZC for each patient across all channels was computed.

The Lempel−Ziv complexity measure (LZ) was used to quantify the complexity of CCEP evoked responses[59]. The LZ method is a robust information-theoretic measure based on universal lossless data compression, and the most popular algorithmic complexity estimator of the Kolmogorov class. The complexity of a one-dimensional signal is estimated by quantifying the number of unique patterns contained within that signal.

CCEPs were band-passed from 0.5 to 10 Hz using a 4th order Butterworth notch filter. Scalp CCEPs were converted to one-dimensional binary signals using the median method. The LZ complexity method was then calculated[59]. The output of the method for computing LZ complexity was then normalized based on the signal sequence length according to:

$$C_{\text{norm}} = \frac{C_{\text{raw}}}{\left[\frac{n}{(n)}\right]} \qquad (1)$$

where, $C_{\text{norm}}$ is the normalized LZ complexity, $C_{\text{raw}}$ is the non-normalized complexity, and $n$ is the signal length.

**Susceptibility index**. We hypothesize that the susceptibility of cortical networks to external perturbations is due to a lack of strong internal dynamics. Thus, the susceptibility is an indication of the poor functional state of the cortical networks. We measured the susceptibility (δ) of the ongoing cortical LFP to external stimulation by quantifying the similarity of the LFP waveform in response to single pulses of stimulation (Fig. 2d, 1 s) and trains of stimulation (Supplementary Fig. 4, 30 s, low-passed LFP signals, <0.1 Hz) across trials[60,61]. This index quantifies the similarity of LFP responses by comparing the variance of the mean of the responses across trials normalized by the average variances of each trial[60].

In Supplementary Fig. 4, the susceptibility index was used to quantify the temporal stability of 30 s LFPs (initial ten seconds that include the stimulation block followed by twenty seconds with no stimulation) across all trials. The susceptibility index is bounded from 0 to 1, and larger values (~1) indicate a highly stable and deterministic response to external stimulation.

**Phase space reconstruction**. We reconstructed the underlying dynamical systems of the cortical networks following sTBI in the phase space using delay embedding methodology. The phase space represents the trajectory of states that the system visits at any time point. Delay embedding theorem enables us to reconstruct this phase space via a single time-series ($X_j = 1, .., N$)[62]. Here, we used depth cortical recordings from the prefrontal cortex to reconstruct the phase space. The one-dimensional time series can be mapped into the higher dimensional space by plotting it against its delayed version as follows:

$$Y_j = (X_j, X_{j+\tau}, X_{j+2\tau}, \dots, X_{j+(d-1)\tau}) \\ j = 1, 2, \dots, N - (d-1)\tau \qquad (2)$$

Here, $d$ is the embedding dimension and $\tau$ the delay. The time delay was estimated based on the signal autocorrelation; the threshold was drawn when the auto-correlation dropped under zero. The minimum sufficient embedding dimension was determined based on the FNN algorithm[36]. The FNN methodology determines the minimum sufficient embedding dimension for $Y_j$ to fully unfold the underlying attractor while preserving its topological characteristics. Under this embedding dimension, all points on the trajectory will be *true* neighbors. FNN determines $d_E$ (minimum needed embedding dimension) by comparing the Euclidean distance for each point in the time series in the $d$-dimensional phase space against a pre-defined distance threshold ($R^{\text{threshold}}$) iteratively increasing $d$ to $d + 1$. At $d = d_E$, the nearest neighbors have a distance less than the threshold, and a further increase in $d$ does not change this distance.

**Principal component analysis (PCA)**. We performed PCA, a widely used dimensionality reduction methodology, to correlate the dimensionality of the recordings to the level of consciousness. We used a MATLAB built-in PCA function to compute the principal components, which are the eigenvectors of the covariance matrix of the dataset. Where the eigenvalues describe how much of the variance in the data is explained by the respective eigenvectors. We performed PCA analysis on the combined scalp ($n = 17$) and depth electrode channels ($n = 8$) on a 5 min stimulation-free period ($n = 25$, Supplementary Fig. 6). These recordings were subject to bipolar re-referencing, normalization, and band-pass filtering (0.5−30 Hz).

**Statistics and reproducibility**. This study included five TBI patients with depth electrophysiological recordings and one control. Statistical analyses were performed on complexity levels across patients using IBM SPSS. CCEPs were delivered in trains of ten pulses separated by one second; typically, each session included ten stimulation trains. Depending on the availability of patients, we recorded between 2 and 6 sessions for each patient. Thus, each patient has between CCEP 200−600 trials. The examples of average CCEP over 100 trials (one session) in the patients with and without thalamic injury are shown in Fig. 2a. To examine statistical differences between the CCEPs waveforms, statistical analyses were performed on complexity levels across patients using IBM SPSS. First, we calculated zero crossing and LZ complexity measures at all the EEG contacts for each trial across patients. Then, we grouped these complexity measures into two categories based on whether patients had a thalamic injury or not. Finally, we used a two-tailed t-test to determine if there was a significant difference in the mean values of these groups. Our results show that the group with thalamic injury had a significantly lower complexity measured with zero-crossing and LZ complexity ($P < 0.01$, two-tailed T-test).

**Network structure for modeling FPN**. A simplified and reduced thalamocortical model has been constructed consisting of three populations of LIF neurons representing frontal and parietal cortical populations and a thalamic population. The cortical populations each include 50 excitatory and inhibitory neurons. The excitatory neurons in each cortical population are connected to all other neurons within their respective cortical population. Further, the excitatory cells in each cortical population project to and receive feedback from their local inhibitory cells, and all the cells within the two cortical populations receive excitatory thalamic inputs. Finally, the synaptic gain of the local and long-distance cortical connections is additionally controlled by the state of thalamic activity via the parameters **α** and **β**, respectively (Fig. 4a).

**Leaky-integrate-and-fire**. The simulated neuronal network is constructed using the LIF model. The LIF neuron integrates all the incoming signals and spikes if the voltage across its membrane reaches a certain threshold ($V_{\text{th}} = -50$ mV). After the spike, the membrane voltage is reset to the resting potential (−65 mV) and it cannot spike for a refractory period of $\tau_{\text{ref}} \sim 1$ ms. The voltage across the LIF cell membrane evolves as follows:

$$C\frac{dV^j}{dt} = -g(V^j - E_L) + I^j_{\text{ext.}} + I^j_{\text{syn.}} + I^j_{\text{stim.}} \qquad (3)$$

Here, $C$ is the membrane capacitance (1 nF), $E_L$ is the resting potential of the neuron, $V^j$ is the voltage across the $j$th neuron membrane, and $g$ is the membrane conductance ($1/R$, where $R = 10$ [$10^6$ $\Omega$]). The neuron's time constant, $\tau$, is equal to $\tau = RC$. Here, $I_{\text{ext}}$ is the driving current for each population of cells (cortical, thalamic), and $I^j_{\text{stim.}}$ is the stimulation pulse that resembles the stimulation delivered by the depth electrode to the $j$th neuron. The synaptic current received by the $j$th cortical neuron is represented by $I^j_{\text{syn.}}$ and is calculated using the following equation, Eq. (4):

$$I^j_{\text{syn.}} = \alpha W^L \sum_j A^L_{i,j} (H(t - t_s - t_d) - H(t - t_s - t_d - T)) \\ + \beta W^D \sum_k A^D_{k,j} (H(t - t_s - t_d) - H(t - t_s - t_d - T)) \\ + W^{\text{Th}} \sum_h A^{\text{Th}}_{h,j} (H(t - t_s - t_d) - H(t - t_s - t_d - T)) \\ - W^{\text{inh}} \sum_n A^{\text{inh}}_{n,j} \left(e^{\frac{1-(t-t_s-t_d)}{Q}}\right) \qquad (4)$$

where the meaning of the symbols is as follows. According to the equation, each cortical cell receives three different synaptic currents: (1) local excitatory and inhibitory, (2) long-distance cortical, and (3) thalamic. The first and the second terms in the equation correspond to the first two synaptic currents: where $A^L_{i,j}$ and $A^D_{k,j}$ matrices are local and long-distance connectivity matrices representing each synaptic connection in the cortical network. Zero entries of these matrices stand for no connection between neurons $i$ and $j$, and entries of one represent the existence of a connection. We assumed all-to-all connectivity for connections within each cortical population and a partial connectivity across populations (each neuron establishes long-distance connections within the radius of fifteen). The strength of these connections is determined with $W^L$ (0.001) and $W^D$ (0.004), which are the local and long-distance synaptic weights, respectively. We assumed 3 ms synaptic delay for all connections ($t_d$). The $H$ operator indicates the Heaviside function,

which controls the duration of the incoming square pulse ($T$) following the presynaptic cell spike ($t_s$).

In this model, the thalamic activity controls the gain of local and long-distance connections through the $\alpha$ and $\beta$ coefficients, respectively. Here, the coma state was characterized by reduced thalamic activity. The terms $\alpha$ and $\beta$ are greater in the awake state in comparison to the coma state ($\alpha_{awake}/\alpha_{coma} = 2$, $\beta_{awake}/\beta_{coma} = 1.5$). The third term corresponds to the thalamic excitatory input to the cortical cells where $W^{Th}$ is the synaptic weight for the *driving* thalamic inputs to the cortex $W^{Th}$ (0.01), and $A^{Th}$ is the connectivity matrix between thalamic and cortical cells (we assumed all-to-all connectivity, $A^{Th}$ is a unity matrix). In the simulated coma state, the firing rate of thalamic cells and thus *driving* input from the thalamus to cortical networks was reduced. This was implemented by: (1) increasing the time constant of the thalamic cells in coma ($\tau = 100$ ms) compared to the awake state ($\tau = 10$ ms), (2) reducing the noisiness of external current (lowering probability [*0.5] and magnitude [*0.025] of the excitatory noise input) relative to the deterministic external current to the thalamic cells in coma ($I^j_{ext.} = 1.5$ nA, in awake $I^j_{ext.} = 1.49$ nA). Together, in this model, thalamic activity controls cortical dynamics in coma and consciousness by adjusting the excitatory *driving* input to the cortex, and the gain of local and long-distance cortical connections.

In addition to the aforementioned excitatory currents, each cortical cell also receives inhibitory input from local inhibitory cells—the fourth term in Eq. (4). The inhibitory cells control the high excitation level in this model and only spike when they receive synchronous excitatory inputs. After spiking, the inhibitory cells inject the local cortical cells with a long-lasting negative current with a synaptic delay of $t_d = 3$ ms that decays exponentially with time. The LFP was obtained by the summation of the excitatory and inhibitory synaptic currents at each cortical area.

**Formation of functional local and long-distance cell assemblies**. In this model, we also investigated the functional state of cell assemblies in the coma and awake states. As mentioned previously, each simulated neuron in the cortical areas is locally connected to the rest of the local cells with $W^L$. To investigate the activity of the cortical representations in the coma and awake states, we embed a cluster of neurons by increasing the $W^L$ to $W^{L'}$ (*4) within a subpopulation of these neurons (neurons 1−20). Hypothetically, this cluster could potentially encode certain aspects of a task or memory. Figure 4 shows that this cluster can be functionally activated when receiving further amplification from the thalamus. In the coma state, without thalamic amplification, the connections within this cluster are too weak to be functionally distinguished. However, in the awake state, the general thalamic amplification of connections and selective amplification within this cluster (*3) turns the cluster ON. As a result, the activity within the cluster becomes remarkably synchronous, distinguishing it from the rest of the cells.

**Reporting summary**. Further information on research design is available in the Nature Research Reporting Summary linked to this article.

## Data availability

All source data for our figures are included as Supplementary Data files. Original recordings and images are available upon reasonable request.

## Code availability

All codes used in this work are available upon reasonable request.

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

## Acknowledgements

We thank Susan Fiore, Dr. Il Memming Park, Josue Nasser, Dr. Ramin Parsey, Dr. Christine DeLorenzo, Dr. Raphael Davis, Dr. Elliot H. Smith, Dr. Justine Liang, Dr. Helen Hsieh, Bradley Ashcroft, and Dr. John Anthony Servider for their helpful comments and discussions. We also thank the Neurosurgery Department, Stony Brook EEG center, and Stony Brook University Hospital for providing support. This work was supported by the National Science Foundation through the Growing Convergence Research (NSF Award 2021002), a Targeted Research Opportunity Program FUSION award (63845) from the Renaissance School of Medicine at Stony Brook University, as well as seed grant funding from the Office of the Vice President for Research at Stony Brook University.

## Author contributions

S.M. contributed to conceptualization, data curation, formal analysis, computational modeling, investigation, methodology, software, validation, visualization, original draft writing, review and editing, and supervision of the research. A.F. was involved with data curation and investigation. J.A. contributed to data curation, draft review and editing, and visualization. P.L.S and T.D. helped with imaging analysis. J.R.S. was involved in visualization and draft review and editing. N.J.W., H.S., and G.F. contributed to draft review and editing. P.D. contributed to conceptualization and draft review and editing. CBM was involved in conceptualization, investigation, project administration, resources, supervision of the research, original draft writing, and review and editing.

## Competing interests

The authors declare no competing interests.
