## [Peer Review File · Communications Biology]

Electrocorticography reveals thalamic control of cortical dynamics following traumatic brain injuryReviewers' comments:

Reviewer #1 (Remarks to the Author):

This manuscript looks into the complex cortical dynamics associated with returning to consciousness in severe TBI patients by measuring local field potentials using invasive electrophysiological recordings. The article also examines these dynamics among patients with thalamic injuries and those without.

Although the sample size is small ($n=5$), the authors provide convincing evidence that thalamic injuries result in a dysfunctional state with simple, low variability and lesser dimensionality in coma state compared to the conscious state. Furthermore, the authors built a model using a simplified network of leaky-integrate-and-fire neurons where they were able to replicate clinical findings that thalamic injuries encompasses delta-band, low complexity local field potentials and low-dimensionality.

This is a very well written manuscript. The data obtained from the patient is of high-quality and the analysis is quite novel demonstrating the cortical dynamics from coma to a conscious state. Although the 'n' used in this study is very small, this is a very difficult population to get data on especially on those that have thalamic injuries.

Some minor corrections are needed.

1. Page 6, Line 93 and 95: I believe the authors really want to refer to Table 1 and not Table S1.
2. Page 19, Line 361: I believe the authors really want refer to Table 1 and not Table S1. Please check.

Reviewer #2 (Remarks to the Author):

The authors of this investigation took advantage of a rare and novel clinical opportunity regarding the implementation of frontal depth electrodes (ten-contact stereotactic depth electrode) spanning from DLPFC to ACC in 5 patients under an epileptic and unconscious state. Depth electrodes were able to stimulate and record local field potential (LFP) in the frontal cortex. Thus, these frontal depth electrodes have provided them the opportunity to track LFP in the anterior brain during consciousness restoration. Besides, they've used single pulse stimulation to measure by PCI the complexity and variability of EEG cortico-cortical evoked potentials. Finally, they developed a computational model to simulate cortico-thalamic neural interactions (in silico) to infer the role of the thalamus depending on its functional and morphological state (damaged vs non-damaged thalamus).

Undoubtedly, this research includes several novel knowledge and it also replicates other important results from the realm of consciousness studies. According to authors, the main result or novelty is the combination of electrocorticography (stimulation and recordings), behavioral and neuroimaging data to infer the role of the thalamus in the cortex during consciousness restoration. Other relevant results were: (1) deep LFP recordings of the PFC "awakening" during the recovery of consciousness, (2) data pointing out that the recovery of consciousness is related to the presence of faster theta, alpha and beta, being theta oscillations in ACC and beta oscillations in PFC as hallmarks (although this result has been already obtained from scalp electrodes, in my knowledge, this is the first time that it is also collected from deep electrodes during the recovery of consciousness) and (3) the use of the PCI to measure the frontal response to single pulses of electrical stimulation to be able to reconstruct a cortical phase space with the capacity to discriminate between thalamic and non-thalamic brain damage patients. All this information was compared with a silico model of the functioning of the FPN to infer thalamus' role in thalamo-cortical dynamics. The main conclusion from this computational model was that the thalamus is an essential agent or structure in the modulation of cortico-cortical networks because it enables the formation of neuronal ensembles required for the content of consciousness. Although conclusions are a knowledge that neuroscientists already know, I find the methodology and some of the results enough relevant to consider the manuscript significant in consciousness research. Anyway, I have some concerns that I would like to ask to the authors.

In addition, the references that I use in some points of this review are to make clear my

suggestions. It is not necessary to include them in the manuscript if the authors find better options or if they consider they are not necessary.

Introduction

I like the introduction, but I think it needs to be more focused on the main topics of the manuscript: consciousness from a clinical and neurofunctional point of view, conscious vs unconscious (is it related with disorders of consciousness?), consciousness restoration in brain damage (role of thalamus, frontal lobe and thalamo-cortical connectivity in the restoration of consciousness). As a clinical reader, I would like to know what has been done before and what did you add up to.

Methodology

- In the Supplementary Fig.6, I can see some neuroimages from patients, but I miss a brief neurological description of them: type of lesion, clinical phase (acute, subacute...) of the patients when MRIs were taken. I consider this information relevant in order to know from what kind of clinical condition (type of lesion + GCS) they have been awaked.
- Throughout the manuscript, authors allude repeatedly to the unconscious state of the patients. Could they describe what is an unconscious state and if all patients were in the same unconscious state? I mean, I can see that mostly all subjects had a 0 score in the CRS-R, but according to the CRS-R, this patient could be in an unresponsive wakefulness syndrome/vegetative state. Why did the authors refer to the patients or to some patients as if they are in coma? Which criteria did they use? This data, jointly other physiopathological parameters, are important in the prognosis of consciousness recovery.
- This is a personal fixation, but in a neurological context I prefer to use the term nonconscious than unconscious, because "unconscious" reminds me to some extent to neuropsychanalytic concepts.
- When were the CRS-R scores taken? In the admission of the Intensive Unit Care or in the onset of the study? Was the CRS-R administered just one or several times?
- I would also include a timeline with some data about the span of time between the injury -- admission -- the implementation of the depth electrodes -- the recovery of consciousness. I find this information very relevant in order to track about the clinical progression of the patients.
- Why did the authors choose to simulate FPN instead of the Default Mode Network (DMN)? I guess because dorsolateral PFC and ACC is related to FPN. But the ACC is also related to the DMN and it has been associated with the restoration of consciousness too (see references below). Please, could the authors discuss or note why they have selected to simulate the FPN instead of the DMN?

References about the relationship between ACC and DMN

Wenyu Tu , Zilu Ma , Yuncong Ma , David Dopfel , Nanyin Zhang. (2021). Suppressing Anterior Cingulate Cortex Modulates Default Mode Network and Behavior in Awake Rats. *Cereb Cortex*. 31:312-323. doi: 10.1093/cercor/bhaa227.

Levar N, Van Doesum TJ, Denys D, Van Wingen GA. Anterior cingulate GABA and glutamate concentrations are associated with resting-state network connectivity. *Sci Rep*. 2019 Feb 14;9(1):2116. doi: 10.1038/s41598-018-38078-1. PMID: 30765822; PMCID: PMC6375948.

Greicius MD, Krasnow B, Reiss AL, Menon V. Functional connectivity in the resting brain: a network analysis of the default mode hypothesis. *Proc Natl Acad Sci U S A*. 2003 Jan 7;100(1):253-8. doi:

10.1073/pnas.0135058100. Epub 2002 Dec 27. PMID: 12506194; PMCID: PMC140943.

References about the relationship between DMN and the restoration of consciousness

Threlkeld ZD, Bodien YG, Rosenthal ES, Giacino JT, Nieto-Castanon A, Wu O, Whitfield-Gabrieli S, Edlow BL. Functional networks reemerge during recovery of consciousness after acute severe traumatic brain injury. *Cortex*. 2018 Sep;106:299-308. doi: 10.1016/j.cortex.2018.05.004. Epub 2018 May 12. PMID: 29871771; PMCID: PMC6120794.

Fingelkurts AA, Fingelkurts AA, Bagnato S, Boccagni C, Galardi G. The Chief Role of Frontal Operational Module of the Brain Default Mode Network in the Potential Recovery of Consciousness from the Vegetative State: A Preliminary Comparison of Three Case Reports. *Open Neuroimaging J*. 2016 May 13;10:41-51. doi: 10.2174/18744440001610010041. PMID: 27347264; PMCID: PMC4894863.

Mäki-Marttunen V, Castro M, Olmos L, Leiguarda R, Villarreal M. Modulation of the default-mode network and the attentional network by self-referential processes in patients with disorder of consciousness. *Neuropsychologia*. 2016 Feb;82:149-160. doi: 10.1016/j.neuropsychologia.2016.01.022. Epub 2016 Jan 18. PMID: 26796715.

- I barely find information about the control subject. Could the authors introduce clinically and experimentally the control subject?

- Although the authors provide some information about "Electrophysiological recordings and stimulations" in the Fig.3 and in "Materials and Methods" section, I still miss some relevant clinical information to understand the results of the experiments for each patient: number of days of stimulation (single day or several days/weeks), number of days between the injury and the onset of the stimulation, initial and final CRS-R for each session of stimulation...

- There is a lack of information about the clinical status of the subjects and the procedure of the experiment. Besides, the information that is included in the manuscript is scattered throughout the main text and the supplementary data, and sometimes even it is very vague ("over days of recording" in page 6, line 116-117). So, I highly recommend to the authors to include a figure to show and condense all the experimental procedure (stimulation characteristics) and clinical information (CRS-R, times, dates... from admission to the recovery of consciousness) for each subject in the "Material and Methods" section. From a clinical point of view there are a lot of variables that incide on the recovery of consciousness and if this information is not explicitly in the manuscript, readers could speculate about the reason why they recover consciousness. The day of admission to the hospital would be the day 0, and other relevant landmarks would be the implementation of the electrodes, how many sessions did they have of electrical stimulation, CRS-R before and after each stimulation session, when did the patients wake up... and other variables that have been suggested above.

- Why were the electrodes placed on the left hemisphere in subject 2? I guess it is because of the focus of the seizures, but please, could authors provide a clinical explanation?

- The CRS-R score shown in page 37, line 114 (subject 1, CRS-R 7-10) does not match with the CRS-R score shown in the Table 1 (subject 1, CRS-R 2-10). Besides, in the Supplementary Figure 1A subject 1 has a 7-10 CRS-R score in day 1 and 2... I'm really confused with these numbers. Could authors clarify this?

- In table 1, I think authors forgot a CRS-R score of subject 4. Furthermore, here there is an example of lack of clinical information. In table 1, CRS-R scores have a "initial" and "final" but of what? I guess stimulation session/s, but one single stimulation session? multiple sessions? I could not find this information along the text.

Results

- In figure 2D there are some "clinical" states in the "x" axis... could the authors explain objectively what exactly do they mean? I mean, "Fast recovery" what does it mean? They recover consciousness quickly since the admission? since the stimulation session/sessions? How many days are the difference between a fast recovery and delayed recovery?... Please, pay attention of this kind of details that are very relevant for the clinicians.

Discusión

- The authors state in the discussion "These data have important implications for coma therapeutics" ... Could the authors discuss this statement deeper? Please make some statements about its possible implications not only in coma, but in the disorders of consciousness too. I think, in the clinical field, there are yet some challenges to determine which severe brain damage patients show a better prognosis in order to be available for some therapeutics options such as sensory stimulation or direct current stimulation programs.

Azabou E, Navarro V, Kubis N, Gavaret M, Heming N, Cariou A, Annane D, Lofaso F, Naccache L, Sharshar T. Value and mechanisms of EEG reactivity in the prognosis of patients with impaired consciousness: a systematic review. *Crit Care*. 2018 Aug 2;22(1):184.

- Could the authors provide some discussion about why the patient with left unilateral damage in the thalamus has restored consciousness whereas the patient with bilateral thalamic damage did not? Did you have other clinical information about the cognitive status of the subject 4 after the recovery of consciousness? Do you think patients with similar lesions on the right thalamus are able to recover consciousness as your patient did? It could be interesting some sentences discussing why the patient with intact left thalamus could recover consciousness whereas the patient with bilateral thalamic lesions couldn't.

- Lines from 134 to 137 indicate that most single-pulse stimulation in the study was administered in the ACC. So, I have some questions:

- You've mentioned most of the stimulations were administered in the ACC. When they were not? Was it protocolized? I know there was a protocol (I saw the reference), but could the authors share the principal points of that protocol in the manuscript?
- Did the authors think that the stimulation of the ACC and dlPFC could be related to the awake of the patients? I can't think about this possibility because of the lack of clinical information. I mean, if the stimulation was close to the admission time, the restoration of consciousness could be related with the natural neurophysiological course in the clinical progression of the patient. Instead, if the stimulation was implemented far away from the admission and clinicians stimulated electrically several times, maybe, those stimulations could affect the restoration of consciousness. This is why I insist in the inclusion of more specific experimental and clinical data. We need it to understand what happened from a clinical point of view.

Thibaut A, Bruno MA, Ledoux D, Demertzi A, Laureys S. tDCS in patients with disorders of consciousness: sham-controlled randomized double-blind study. *Neurology*. 2014 Apr 1;82(13):1112-8. doi: 10.1212/WNL.0000000000000260. Epub 2014 Feb 26. PMID: 24574549.

Angelakis E, Liouta E, Andreadis N, Korfiatis S, Ktonas P, Stranjalis G, Sakas DE. Transcranial direct current stimulation effects in disorders of consciousness. *Arch Phys Med Rehabil*. 2014 Feb;95(2):283-9. doi: 10.1016/j.apmr.2013.09.002. Epub 2013 Sep 11. PMID: 24035769.

Reviewer #3 (Remarks to the Author):

The authors recorded LFP/EEG and stimulated depth electrodes implanted in the prefrontal cortex (PFC) and the anterior cingulate cortex (ACC) for seizure monitoring after sTBI. They used computational modeling to gain insight into the role of the thalamus in shaping the functional state of the cortex in the context of consciousness recovery. Depth recordings in sTBI patients revealed complex dynamics if the thalamic projections were intact, and simple (attractor-like) dynamics if they were impaired.

This work was well-organized and dealt with directly measured signals of PFC and ACC. However, one of the weakest points is the lack of direct evidence (or measurement) for the thalamic engagement. To argue the important involvement of the thalamus in regard to the consciousness, this paper employed only an in-silico model rather than empirical data directly from the thalamus.

The authors used primarily the low-dimensional attractor analysis in a reconstructed phase space. This approach is frequently used in physics. Moreover, their in-silico modeling lacks many of the biological details of the thalamocortical circuit. The conventional EEG/LFP analyses (e.g., spectral analysis or correlation analysis or cross-frequency coupling) are additionally necessary to reinforce their arguments. The following are other miscellaneous points:

1. In lines 62-64, the reference #22 was published in 2012, which is not recent. However, that sentence reads "More recently, ...".
2. In Fig. 1A, the exact locations of PFC and ACC are not clear. Providing their MNI coordinates would be helpful.
3. In Fig. 2A, the evoked activity for No-recovery condition showed higher deflections in the right hemisphere compared to in the left hemisphere. What could be a plausible explanation for these lateralized responses?
4. The discussion section could be further elaborated, particularly if focused on their observations.
5. In the Method section, the impedance of electrodes and the criteria for artifact-rejection should be reported.
6. The statistical analytic method was not clearly described.

May 21, 2021

Dear colleagues,

We thank the reviewers for their detailed attention to our manuscript. We address criticisms line-by-line as follows:

Reviewer #1 (Remarks to the Author):

This manuscript looks into the complex cortical dynamics associated with returning to consciousness in severe TBI patients by measuring local field potentials using invasive electrophysiological recordings. The article also examines these dynamics among patients with thalamic injuries and those without.

Although the sample size is small ($n=5$), the authors provide convincing evidence that thalamic injuries result in a dysfunctional state with simple, low variability and lesser dimensionality in coma state compared to the conscious state. Furthermore, the authors built a model using a simplified network of leaky-integrate-and-fire neurons where they were able to replicate clinical findings that thalamic injuries encompasses delta-band, low complexity local field potentials and low-dimensionality.

This is a very well written manuscript. The data obtained from the patient is of high-quality and the analysis is quite novel demonstrating the cortical dynamics from coma to a conscious state. Although the 'n' used in this study is very small, this is a very difficult population to get data on especially on those that have thalamic injuries.

Some minor corrections are needed.

1. Page 6, Line 93 and 95: I believe the authors really want to refer to Table 1 and not Table S1.

We thank the reviewer for their comments. This has been fixed.

2. Page 19, Line 361: I believe the authors really want to refer to Table 1 and not Table S1. Please check.

We have expanded the description of clinical histories based on reviewer feedback. This is now referred to in the supplementary text. We thank the reviewer for calling our attention to this detail.

Reviewer #2 (Remarks to the Author):

The authors of this investigation took advantage of a rare and novel clinical opportunity regarding the implementation of frontal depth electrodes (ten-contact stereotactic depth electrode) spanning from DLPFC to ACC in 5 patients under an epileptic and unconscious state. Depth electrodes were able to stimulate and record local field potential (LFP) in the frontal cortex. Thus, these frontal depth electrodes have provided them the opportunity to track LFP in the anterior brain during consciousness restoration. Besides, they've used single

pulse stimulation to measure by PCI the complexity and variability of EEG cortico-cortical evoked potentials. Finally, they developed a computational model to simulate cortico-thalamic neural interactions (in silico) to infer the role of the thalamus depending on its functional and morphological state (damaged vs non-damaged thalamus).

Undoubtedly, this research includes several novel knowledge and it also replicates other important results from the realm of consciousness studies. According to authors, the main result or novelty is the combination of electrocorticography (stimulation and recordings), behavioral and neuroimaging data to infer the role of the thalamus in the cortex during consciousness restoration. Other relevant results were: (1) deep LFP recordings of the PFC "awakening" during the recovery of consciousness, (2) data pointing out that the recovery of consciousness is related to the presence of faster theta, alpha and beta, being theta oscillations in ACC and beta oscillations in PFC as hallmarks (although this result has been already obtained from scalp electrodes, in my knowledge, this is the first time that it is also collected from deep electrodes during the recovery of consciousness) and (3) the use of the PCI to measure the frontal response to single pulses of electrical stimulation to be able to reconstruct a cortical phase space with the capacity to discriminate between thalamic and non-thalamic brain damage patients. All this information was compared with a silico model of the functioning of the FPN to infer thalamus' role in thalamo-cortical dynamics. The main conclusion from this computational model was that the thalamus is an essential agent or structure in the modulation of cortico-cortical networks because it enables the formation of neuronal ensembles required for the content of consciousness. Although conclusions are a knowledge that neuroscientists already know, I find the methodology and some of the results enough relevant to consider the manuscript significant in consciousness research. Anyway, I have some concerns that I would like to ask to the authors.

In addition, the references that I use in some points of this review are to make clear my suggestions. It is not necessary to include them in the manuscript if the authors find better options or if they consider they are not necessary.

We thank the reviewer for his or her positive evaluation of our work.

Introduction

I like the introduction, but I think it need to be more focused on the main topics of the manuscript: consciousness from a clinical and neurofunctional point of view, conscious vs unconscious (is it related with disorders of consciousness?), consciousness restoration in brain damage (role of thalamus, frontal lobe and thalamo-cortical connectivity in the restoration of consciousness). As a clinical reader, I would like to know what has been done before and what did you add up to.

We thank the reviewer for their suggestion. We have added references to clinical trials of thalamic stimulation in TBI, which have both mechanistic and clinical significance (page 5, lines 72-74).

Methodology

- In the Supplementary Fig.6, I can see some neuroimages from patients, but I miss a brief neurological description of them: type of lesion, clinical phase (acute, subacute...) of the patients when MRIs were taken. I consider this information relevant in order to know from what kind of clinical condition (type of lesion + GCS) they have been awaked.

We have provided brief clinical summaries as supplementary text and Supplementary Figure 7.

- Throughout the manuscript, authors allude repeatedly to the unconscious state of the patients. Could they describe what is an unconscious state and if all patients were in the same unconscious state? I mean, I can see that mostly all subjects had a 0 score in the CRS-R, but according to the CRS-R, this patient could be in an

unresponsive wakefulness syndrome/vegetative state. Why did the authors refer to the patients or to some patients as if they are in coma? Which criteria did they use? This data, jointly other physiopathological parameters, are important in the prognosis of consciousness recovery.

We used GCS < 8 as a broad screen, but patients were selected on an individual basis. We agree with the reviewer and details of clinical histories are in the new supplementary text.

- This is a personal fixation, but in a neurological context I prefer to use the term nonconscious than unconscious, because "unconscious" reminds me to some extent to neuropsychanalytic concepts.

We see the reviewer's point, but we worry that "nonconscious" would sound strange to the general reader of Communications Biology (it sounds like it could be confused with "incapable of consciousness," i.e., a nonconscious porcelain doll). We have rephrased specific instances to avoid "unconscious," in favor of "comatose." In one case, we thought the text read better left alone (page 4, line 47).

- When were the CRS-R scores taken? In the admission of the Intensive Unit Care or in the onset of the study? Was the CRS-R administered just one or several times?

CRS-R was assessed subsequent to study entry. Please see Supplementary Figure 7 and the accompanying supplementary text.

- I would also include a timeline with some data about the span of time between the injury -- admission -- the implementation of the depth electrodes -- the recovery of consciousness. I find this information very relevant in order to track about the clinical progression of the patients.

We thank the reviewer for this important criticism. We have included this information in Supplementary Figure 7 and accompanying supplementary text.

- Why did the authors choose to simulate FPN instead of the Default Mode Network (DMN)? I guess because dorsolateral PFC and ACC is related to FPN. But the ACC is also related to the DMN and it has been associated with the restoration of consciousness too (see references below). Please, could the authors discuss or note why they have selected to simulate the FPN instead of the DMN?

We thank the reviewer for this important point. The goal of the experiments described was to understand brain circuits needed for goal-directed behavior; by definition, DMN is not involved in behavior. The reviewer is correct that DMN connectivity is closely related to consciousness (as described in the extensive literature he or she cites). But we think that DMN is outside the scope of the present work.

- I barely find information about the control subject. Could the authors introduce clinically and experimentally the control subject?

Additional information about the control subject has been added to the supplementary text:

"The control subject was a 24-year-old woman with refractory epilepsy who underwent an extensive stereo-EEG depth electrode implant with bitemporal and bifrontal coverage, including bilateral electrodes which spanned prefrontal cortex and dorsal anterior cingulate, in an identical anatomical location to the study patients. No seizures were identified during the period of recording, however."

- Although the authors provide some information about "Electrophysiological recordings and stimulations" in the Fig.3 and in "Materials and Methods" section, I still miss some relevant clinical information to understand the results of the experiments for each patient: number of days of stimulation (single day or several days/weeks), number of days between the injury and the onset of the stimulation, initial and final CRS-R for each session of stimulation...

Additional information has been added to Supplementary Figure 7 and the supplementary text.

- There is a lack of information about the clinical status of the subjects and the procedure of the experiment. Besides, the information that is included in the manuscript is scattered throughout the main text and the supplementary data, and sometimes even it is very vague (“over days of recording” in page 6, line 116-117). So, I highly recommend to the authors to include a figure to show and condense all the experimental procedure (stimulation characteristics) and clinical information (CRS-R, times, dates... from admission to the recovery of consciousness) for each subject in the “Material and Methods” section. From a clinical point of view there are a lot of variables that incide on the recovery of consciousness and if this information is not explicitly in the manuscript, readers could speculate about the reason why they recover consciousness. The day of admission to the hospital would be the day 0, and other relevant landmarks would be the implementation of the electrodes, how many sessions did they have of electrical stimulation, CRS-R before and after each stimulation session, when did the patients wake up... and other variables that have been suggested above.

We thank the reviewer for this important criticism. Details have been added to Supplementary Figure 7 and supplementary text.

- Why were the electrodes placed on the left hemisphere in subject 2? I guess it is because of the focus of the seizures, but please, could authors provide a clinical explanation?

The electrode was placed there for concerns related to scalp availability for future procedures. Details have been added to the supplementary text.

- The CRS-R score shown in page 37, line 114 (subject 1, CRS-R 7-10) does not match with the CRS-R score shown in the Table 1 (subject 1, CRS-R 2-10). Besides, in the Supplementary Figure 1A subject 1 has a 7-10 CRS-R score in day 1 and 2... I’m really confused with these numbers. Could authors clarify this?

Details have been added to supplementary figure 7 and supplementary text. We fixed the typo in the Supplementary Figure 1.

- In table 1, I think authors forgot a CRS-R score of subject 4. Furthermore, here there is an example of lack of clinical information. In table 1, CRS-R scores have a "initial" and "final" but of what? I guess stimulation session/s, but one single stimulation session? multiple sessions? I could not find this information along the text.

Details have been added to the Supplementary Figure 7 and supplementary text.

Results

- In figure 2D there are some "clinical" states in the “x” axis... could the authors explain objectively what exactly do they mean? I mean, “Fast recovery” what does it mean? They recover consciousness quickly since the admission? since the stimulation session/sessions? How many days are the difference between a fast recovery and delayed recovery?... Please, pay attention of this kind of details that are very relevant for the clinicians.

We thank the reviewer for their comment. We have clarified these details in the figure legend (page 9, lines 156-158). Subject 2 had a rapid increase in CRS-R, especially over his second week, as shown in supplementary figure 7. Again we thank the reviewer for helping us clarify these important points.

Discusión

- The authors state in the discussion “These data have important implications for coma therapeutics” ... Could the authors discuss this statement deeper? Please make some statements about its possible implications not only in coma, but in the disorders of consciousness too. I think, in the clinical field, there are yet some challenges to determine which severe brain damage patients show a better prognosis in order to be available for some therapeutics options such as sensory stimulation or direct current stimulation programs.

Again we thank the reviewer for helping us clarify these important points. We have spelled out why arousal-only interventions for TBI are unlikely to bring about consciousness in many injured patients (page 16, lines 297-299).

- Could the authors provide some discussion about why the patient with left unilateral damage in the thalamus has restored consciousness whereas the patient with bilateral thalamic damage did not? Did you have other clinical information about the cognitive status of the subject 4 after the recovery of consciousness? Do you think patients with similar lesions on the right thalamus are able to recover consciousness as your patient did? It could be interesting some sentences discussing why the patient with intact left thalamus could recover consciousness whereas the patient with bilateral thalamic lesions couldn't.

The critical role of thalamocortical projections has not been adequately appreciated. We have added to the discussion of this patient (page 17, lines 310-313).

- Lines from 134 to 137 indicate that most single-pulse stimulation in the study was administered in the ACC. So, I have some questions:

- You've mentioned most of the stimulations were administered in the ACC. When they were not? Was it protocolized? I know there was a protocol (I saw the reference), but could the authors share the principal points of that protocol in the manuscript?

We administered single pulses of stimulation both at contacts located in PFC and ACC and we observed stimulation in ACC spreads to most EEG contacts, while stimulation in PFC remains local (Supplementary Figure 2). The stimulation protocol is described in lines 392-398.

- Did the authors think that the stimulation of the ACC and dlPFC could be related to the awake of the patients? I can't think about this possibility because of the lack of clinical information. I mean, if the stimulation was close to the admission time, the restoration of consciousness could be related with the natural neurophysiological course in the clinical progression of the patient. Instead, if the stimulation was implemented far away from the admission and clinicians stimulated electrically several times, maybe, those stimulations could affect the restoration of consciousness. This is why I insist in the inclusion of more specific experimental and clinical data. We need it to understand what happened from a clinical point of view.

Agree with the reviewer. However, the goal of single-pulse stimulation was not to awaken the patients; it was to probe cortico-cortical function and connectivity. While it is possible that stimulation could have changed function in some way, we did not observe changes to the ongoing EEG activity post-stimulation. It is thus unlikely that we could increase arousal in this fashion and needs to be evaluated more carefully when we recruit more patients. We have clarified this point (page 20, lines 397-398).

Reviewer #3 (Remarks to the Author):

The authors recorded LFP/EEG and stimulated depth electrodes implanted in the prefrontal cortex (PFC) and the anterior cingulate cortex (ACC) for seizure monitoring after sTBI. They used computational modeling to gain insight into the role of the thalamus in shaping the functional state of the cortex in the context of consciousness recovery. Depth recordings in sTBI patients revealed complex dynamics if the thalamic projections were intact, and simple (attractor-like) dynamics if they were impaired.

This work was well-organized and dealt with directly measured signals of PFC and ACC. However, one of the weakest points is the lack of direct evidence (or measurement) for the thalamic engagement. To argue the important involvement of the thalamus in regard to the consciousness, this paper employed only an in-silico model rather than empirical data directly from the thalamus.

We thank the reviewer for highlighting this important point. In future studies, we hope to provide direct thalamic recordings from TBI patients, but this was not feasible at this time. We have commented to this effect in the discussion (lines 337-345).

The authors used primarily the low-dimensional attractor analysis in a reconstructed phase space. This approach is frequently used in physics. Moreover, their in-silico modeling lacks many of the biological details of the thalamocortical circuit. The conventional EEG/LFP analyses (e.g., spectral analysis or correlation analysis or cross-frequency coupling) are additionally necessary to reinforce their arguments. The following are other miscellaneous points:

1. In lines 62-64, the reference #22 was published in 2012, which is not recent. However, that sentence reads “More recently, ...”.

We have rephrased this clause. We thank the reviewer for their comment.

2. In Fig. 1A, the exact locations of PFC and ACC are not clear. Providing their MNI coordinates would be helpful.

Postop images were generally CT images so transformation into MNI space is not fully reliable, because of the lack of soft-tissue contrast. Thus we believe it is better to avoid MNI coordinates, so as to avoid the misleading impression of extreme accuracy. PFC was identified as the caudal part of the middle frontal gyrus at the level of the coronal suture, and anterior cingulate was typically targeted 20 mm from the tip of the frontal horn. This has been added to the text (page 20, lines 382-383). Figure 1 shows the locations within PFC and ACC.

3. In Fig. 2A, the evoked activity for No-recovery condition showed higher deflections in the right hemisphere compared to in the left hemisphere. What could be a plausible explanation for these lateralized responses?

Electrodes were right-sided for all patients except for subject 2. The amplitudes are bigger because they are closer to stimulation onset. We have clarified this detail on page 20, lines 377-378 and in the supplementary text.

4. The discussion section could be further elaborated, particularly if focused on their observations.

We have expanded on several points based on criticisms by reviewer #2 (page 16, lines 297-299; page 17, lines 311-315).

5. In the Method section, the impedance of electrodes and the criteria for artifact-rejection should be reported.

Impedances for scalp contacts were within 5-10 k Ω . Impedance was not directly measured on intracranial contacts, but recordings were visually inspected for quality. Artifacts were rejected by FieldTrip toolbox and visual inspection. These details have been added to the text (page 20, lines 385-388).

6. The statistical analytic method was not clearly described.

Descriptions of statistical analytics (T-tests performed, PCA, Lempel-Ziv complexity) are available on pages 23-26, lines 430-495. Additional information about statistics for zero crossing and LZ has been added to the methods.

Again we thank the reviewers for their detailed criticism of our work, and we thank the editor for the opportunity to improve the manuscript.

Sincerely,

Charles Mikell M.D.
Clinical Assistant Professor of Neurosurgery
Co-director, Center for Movement Disorders

Reviewers' comments:

Reviewer #1 (Remarks to the Author):

The authors have done a good job responding to the previous review. Particularly the comments from the second reviewer and the responses to those comments makes the manuscript even better.

Reviewer #2 (Remarks to the Author):

No comments to the editors are needed.

Reviewer #3 (Remarks to the Author):

The authors have not yet addressed my following comments:

"The authors used primarily the low-dimensional attractor analysis in a reconstructed phase space. This approach is frequently used in physics. Moreover, their in-silico modeling lacks many of the biological details of the thalamocortical circuit. The conventional EEG/LFP analyses (e.g., spectral analysis or correlation analysis or cross-frequency coupling) are additionally necessary to reinforce their arguments".

I would consider this manuscript for publication after receiving full responses to each of my concerns.

**Stony Brook
Medicine**

Neurosciences Institute
Department of Neurological Surgery
NY Spine & Brain Surgery, U.F.P.C.

June 23, 2021

Dear colleagues,

We again thank the reviewers for their detailed attention to our manuscript. We address the final two criticisms as follows:

Moreover, their in-silico modeling lacks many of the biological details of the thalamocortical circuit.

We acknowledge that our model is somewhat basic, and lacks the laminar organization of the mammalian thalamocortical circuit. A more detailed model is in development, but we feel it is outside the scope of the present work. Statements to this effect have been added to the Limitations section (lines 348-350).

The conventional EEG/LFP analyses (e.g., spectral analysis or correlation analysis or cross-frequency coupling) are additionally necessary to reinforce their arguments.

Spectral analyses are reported in Figure 1 and Supplementary Figure 1.

We again thank the reviewers for their attention to our manuscript,

Sincerely,

Charles Mikell M.D.
Clinical Assistant Professor of Neurosurgery
Co-director, Center for Movement Disorders

Raphael P. Davis, MD
Professor and Chairman

David Chesler, MD, PhD
Reza Dashti, MD, PhD
Michael Egnor, MD
David Fiorella, MD, PhD
Frederick Gutman, MD
Donald Macron, MD
Charles Mikell, MD
Sima Mofakham, PhD
Courtney Pendleton, MD
Jonathan Raanan, MD

REVIEWERS' COMMENTS:

Reviewer #1 (Remarks to the Author):

The authors have addressed the comments well and I agree that although the in-silico model is somewhat basic it does provide new insights into the role of thalamic inputs associated with consciousness.

From my perspective this manuscript is ready for publication

Reviewer #2 (Remarks to the Author):

Authors has addressed satisfactorily my comments

Reviewer #3 (Remarks to the Author):

My points were addressed.